# Advanced rechargeable aluminium ion battery with a high-quality natural graphite cathode

Di-Yan Wang[1,2,3], Chuan-Yu Wei[1,4], Meng-Chang Lin[3,5], Chun-Jern Pan[3,6], Hung-Lung Chou[7], Hsin-An Chen[4], Ming Gong[3], Yingpeng Wu[3], Chunze Yuan[3], Michael Angell[3], Yu-Ju Hsieh[1], Yu-Hsun Chen[1], Cheng-Yen Wen[4], Chun-Wei Chen[4], Bing-Joe Hwang[6,8], Chia-Chun Chen[1,9] & Hongjie Dai[3]

Recently, interest in aluminium ion batteries with aluminium anodes, graphite cathodes and ionic liquid electrolytes has increased; however, much remains to be done to increase the cathode capacity and to understand details of the anion–graphite intercalation mechanism. Here, an aluminium ion battery cell made using pristine natural graphite flakes achieves a specific capacity of $\sim 110\,\mathrm{mAh\,g^{-1}}$ with Coulombic efficiency $\sim 98\%$, at a current density of $99\,\mathrm{mA\,g^{-1}}$ (0.9 C) with clear discharge voltage plateaus (2.25–2.0 V and 1.9–1.5 V). The cell has a capacity of $60\,\mathrm{mAh\,g^{-1}}$ at 6 C, over 6,000 cycles with Coulombic efficiency $\sim 99\%$. Raman spectroscopy shows two different intercalation processes involving chloroaluminate anions at the two discharging plateaus, while C–Cl bonding on the surface, or edges of natural graphite, is found using X-ray absorption spectroscopy. Finally, theoretical calculations are employed to investigate the intercalation behaviour of choloraluminate anions in the graphite electrode.

[1] Department of Chemistry, National Taiwan Normal University, Taipei 11677, Taiwan. [2] Department of Chemistry, Tunghai University, Taichung 40704, Taiwan. [3] Department of Chemistry, Stanford University, Stanford, California 94305, USA. [4] Department of Materials Science and Engineering, National Taiwan University, Taipei 10617, Taiwan. [5] College of Electrical Engineering and Automation, Shandong University of Science and Technology, Qingdao 266590, China. [6] Department of Chemical Engineering, National Taiwan University of Science and Technology, Taipei 10607, Taiwan. [7] Graduate Institute of Applied Science and Technology, National Taiwan University of Science and Technology, Taipei 10607, Taiwan. [8] National Synchrotron Radiation Research Center (NSRRC), Hsinchu 30076, Taiwan. [9] Institute of Atomic and Molecular Science, Academia Sinica, Taipei 10617, Taiwan. Correspondence and requests for materials should be addressed to B.-J.H. (email: bjh@mail.ntust.edu.tw) or to C.-C.C. (email: cjchen@ntnu.edu.tw) or to H.D. (email: hdai@stanford.edu).

Batteries for electrical energy storage have attracted intense attention due to the steadily increasing demands of mobile and stationary applications[1–4]. Among various batteries, Al-ion batteries (AIBs) are intriguing due to the high gravimetric capacity (2,980 Ah kg$^{-1}$) of the Al anode relative to Li$^+$/Li (3,862 Ah kg$^{-1}$) and Na$^+$/Na (1,166 Ah kg$^{-1}$) that results from the three-electron Al$^{3+}$/Al redox couple[5]. We recently developed a rechargeable AIB with an Al anode that undergoes reversible electrochemical deposition and dissolution at room temperature in an ionic liquid (IL) electrolyte formed by mixing 1-ethyl-3-methylimidazolium chloride ([EMIm]Cl) and aluminium chloride (AlCl$_3$) that produce redox active chloroaluminate anions (AlCl$_4^-$ and Al$_2$Cl$_7^-$)[6,7]. Note that the Al anode was previously paired with a fluorinated graphite as a cathode electrode to afford a capacitor-like device with low discharging voltage of $\sim 1$ V (ref. 8), which differed from our Al–graphite battery with reversible chloroaluminate anion intercalation/de-intercalation redox chemistry[9]. The rate capability can be significantly improved by constructing a highly porous three-dimensional graphitic foam that allows fast ion diffusion/intercalation. A drawback of our previous rechargeable Al battery was the low-specific cathode capacity of $<70$ mAh g$^{-1}$ down to its low C rate, which needed to be improved to match the high capacity of the Al anode side and thus increase the AIB's energy density.

Graphite is an excellent material for hosting a wide range of intercalated guests between the planar graphene sheets[10]. A variety of graphite intercalation reactions and compounds have been studied in the rechargeable battery field[11,12]. The charge storage capacity of graphite is related to the number of ionic intercalants that is limited by their size[10], and is also dependent on the structure and morphology of the graphite materials used as in the case of the graphite anodes used in lithium and sodium ion batteries[13–16]. In addition, graphite materials have been used in aqueous rechargeable zinc/aluminium ion batteries that use zinc as the negative electrode and ultrathin graphite nanosheets as the positive electrode in an aqueous Al$_2$(SO$_4$)$_3$/Zn(CHCOO)$_2$ electrolyte[17]. Our previous AIB that used pyrolytic graphite papers as the cathode material had a relatively low capacity of $\sim 66$ mAh g$^{-1}$. There is a need to investigate graphite materials with a high chloroaluminate anions intercalation capacity with minimal side reactions. Further, it is important to investigate and understand the chloroaluminate anion–graphite intercalation reactions in relation to the electrochemical voltage plateaus, which could hold the key to future battery development.

In this work, we develop a rechargeable AIB using a film of SP-1 natural graphite flakes (NG) with a polyvinylidene fluoride (PVDF) binder as the cathode of a rechargeable AIB (Al/NG cell). Our AIB cell exhibits clear discharge voltage plateaus in the ranges 2.25–2.0 V and 1.9–1.5 V, while the graphite cathode exhibits a much improved specific capacity over pyrolytic graphite up to $\sim 110$ mAh g$^{-1}$ with $\sim 98\%$ Coulombic efficiency (CE) at a current density of $\sim 99$ mA g$^{-1}$. Charge–discharge cycling at a current density of 660 mA g$^{-1}$ shows the cells to be highly stable, with little capacity decay over $>6,000$ cycles at $\sim 99\%$ CE. The reversible structural evolution of the NG particles during charging and discharging is characterized by *in situ* Raman spectroscopy, powder X-ray diffraction, X-ray photoelectron spectroscopy (XPS) and X-ray absorption spectroscopy (XAS). Finally, density functional theory (DFT) calculations are used to investigate and verify the intercalation behaviour of choloraluminate anions in the NG electrodes.

## Results

### Fabrication of NG flake film and performance of the Al/NG cell.

Figure 1 shows a schematic illustration of the process for making a NG flake film suitable for rechargeable Al-ion battery use. First, NG slurry was prepared by mixing SP-1 graphite powder (90%) and PVDF (10%) in *N*-methyl pyrrolidone solvent (Step A). The slurry was cast onto a Cu foil to form a uniform graphite film and dried at 150 °C for 2 h (Step B). The Cu foil was then etched by immersing the sample into the iron chloride (FeCl$_3$) solution (0.4 g ml$^{-1}$; Step C) to form a free-standing NG film. Afterwards, the NG film was rinsed with deionized water to remove the residual FeCl$_3$, and then dried at 200 °C for 3 h (Step D). The resulting NG film ($2 \times 2.3$ cm, Step E of Fig. 1) was bonded to Ni bar current collectors by sandwiching conducting carbon tapes between the NG film and Ni bars to form the NG cathode (see in the photograph of Step E). The average size of the NG flakes was $\sim 90$ μm (200 mesh), and the NG film showed a thickness of $\sim 50$ μm imaged by top and cross-sectional scanning electron microscopy (Fig. 1b,c). The graphite loading was $\sim 4$ mg cm$^{-2}$ in the film.

The Al/NG cell was constructed in a pouch cell (Methods) using a thin Al foil (thickness $\sim 20$ μm) anode, NG film cathode and IL electrolyte with a AlCl$_3$/[EMIm]Cl ratio of $\sim 1.3$. To minimize residual water in the electrolyte, [EMIm]Cl was vacuum dried ($<700$ p.p.m.) before mixing with AlCl$_3$. Figure 2a showed the charge and discharge curves with different current densities from 66 to 792 mA g$^{-1}$ with cutoff voltages of 0.5–2.45 V. Interestingly, we observed two clear discharge voltage plateaus in the ranges 2.25–2.0 V and 1.9–1.5 V. The cell delivered a specific capacity up to 110 mAh g$^{-1}$ based on the mass of graphite (4 mg cm$^{-2}$) at a current density of 66 mA g$^{-1}$ ($\sim 0.6$ C).

We investigated various graphite materials for AIB cathodes and found that in general, NG was superior to synthetic graphite in conferring higher capacities and well-defined voltage plateaus. Two synthetic graphite materials, KS6 and MCMB, both widely used as Li ion battery anode materials, were found to give much lower capacity than NG without clear discharge voltage plateaus (Supplementary Fig. 1). This was attributed to KS6 and MCMB materials with higher surface areas having lower crystallinities and higher defect densities than NG, see Raman and X-ray diffraction measurements (Supplementary Fig. 1). These results indicated that suitability of high crystallinity, low defect density graphite materials for use in high-performance Al-ion batteries; however, in general, we found that graphite materials commonly used for lithium ion batteries performed poorly in AIB.

Figure 2b and Supplementary Fig. 2 shows the rate performance of the Al/NG battery cell. At higher charge–discharge rate or current densities, the cell showed decreased capacity and higher CE. At a high rate of $\sim 6$ C (10 min charge/discharge time under a 660 mA g$^{-1}$ current), the Al/NG battery could still deliver a capacity of 60 mAh g$^{-1}$ and $\sim 99.5\%$ CE. The discharge capacity recovered to 110 mAh g$^{-1}$ when the rate slowed. Charge–discharge cycling at a high rate of 6 C showed the high stability of the AIB without specific capacity decay over 6,000 cycles at a $\sim 99.5\%$ CE (Fig. 2c). Cycling at a lower rate (1.8 C, 198 mA g$^{-1}$) also showed a high stability with a specific capacity of $\sim 100$ mAh g$^{-1}$ over 1,100 cycles (Fig. 2d); while significantly, the charge and discharge curves of the AIB battery recorded at the third and 1,100th cycle were almost identical (inset of Fig. 2d).

Note that we measured cathode-specific capacity versus graphite loading (Supplementary Figs 3 and 4) and found that under a loading of 3–4 mg cm$^{-2}$, a high-specific capacity of 110 mAh g$^{-1}$ was reliably obtained for the graphite cathode. However, with higher graphite loadings, the specific capacity was reduced. Nevertheless, with useful graphite loading levels, AIB cells exhibited high performance in terms of capacity, rate capability and stability.

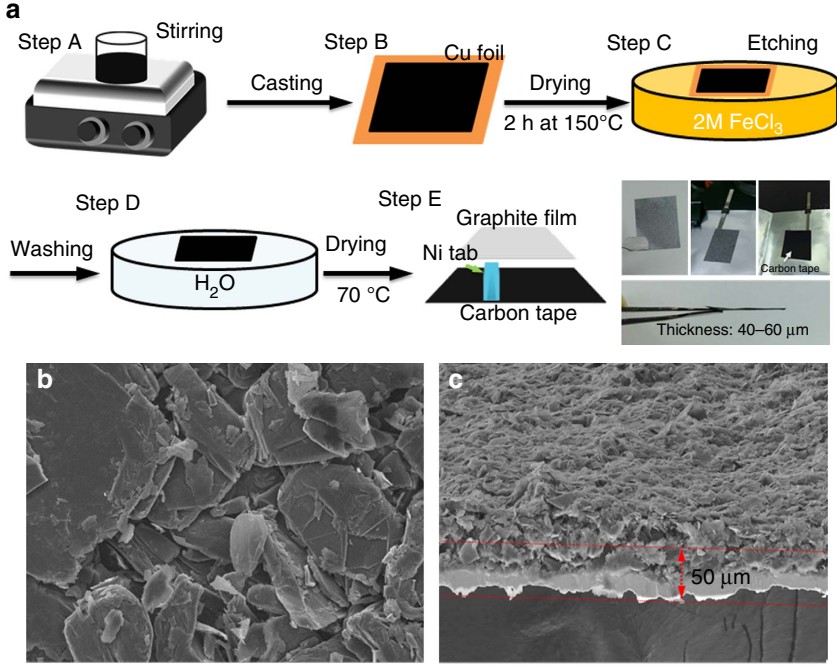

**Figure 1 | Free-standing natural graphite film for aluminium ion battery.** (**a**) Schematic illustration of preparation process of a free-standing natural graphite (NG) film (steps A–D) bonded to a conducting carbon tape current collector (step E). (**b**) The top (scale bars, 10 μm) and (**c**) cross-sectional SEM images of GFF electrode. (scale bars, 100 μm) The thickness of NG film is ∼50 μm. The loading amount of NG is 4 mg cm$^{-2}$, only calculating the weight of graphite without binder.

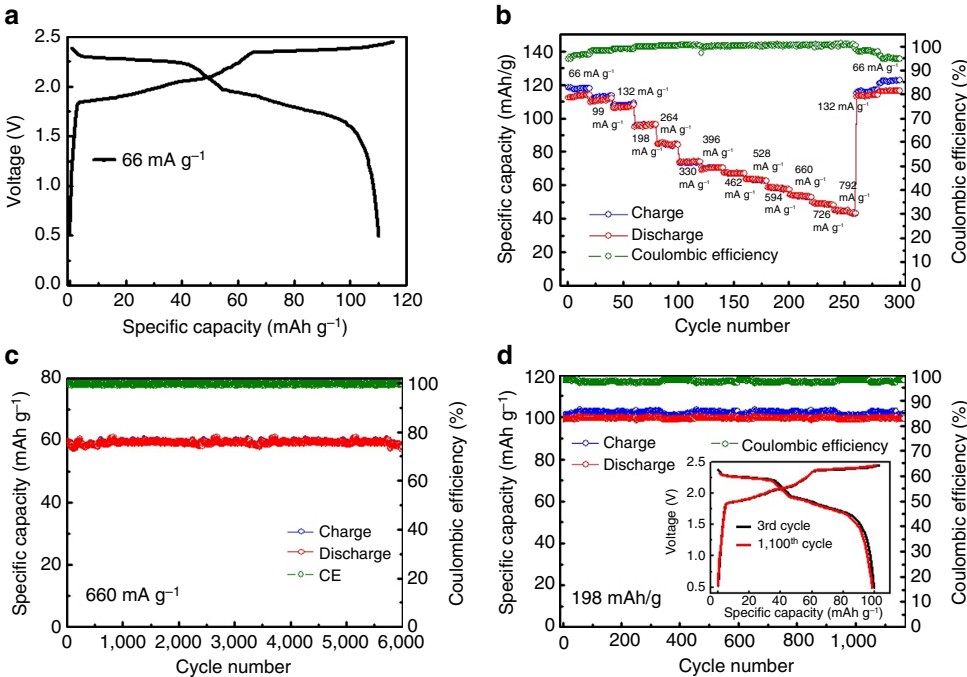

**Figure 2 | Performance of aluminium ion battery.** (**a**) Galvanostatic charge and discharge curves of an Al/NG cell at a current density of 66 mA g$^{-1}$. (**b**) Capacity retention of an Al/NG cell cycled at various current densities. (**c,d**) Long-term stability test of an Al/NG cell at 660 and 198 mA g$^{-1}$, respectively. All capacity of Al/NG battery was recorded between charging and discharging voltages of 0.5 and 2.45 V. Graphite loading mass of all batteries ∼4 mg cm$^{-2}$.

**Structural characterizations of NG during anion intercalation.** Figure 3a shows *ex situ* X-ray diffraction measurements of NG flakes in the cathode of an AIB cell cycled at a constant current density. The (002) peak of pristine NG was observed at $2\theta = 26.05°$. By charging the Al/NG cell to 2.0 V versus Al/Al$^{3+}$ at a constant current of 99 mA g$^{-1}$, the (002) peak completely vanished and gradually split into two new peaks signalling chloroaluminate anion intercalation into NG. When the charging voltage reached 2.45 V, two distinct peaks appeared at $2\theta = 28.4°$ ($d\sim3.14$ Å) and 23.2° ($d\sim3.85$ Å). During discharging, the two

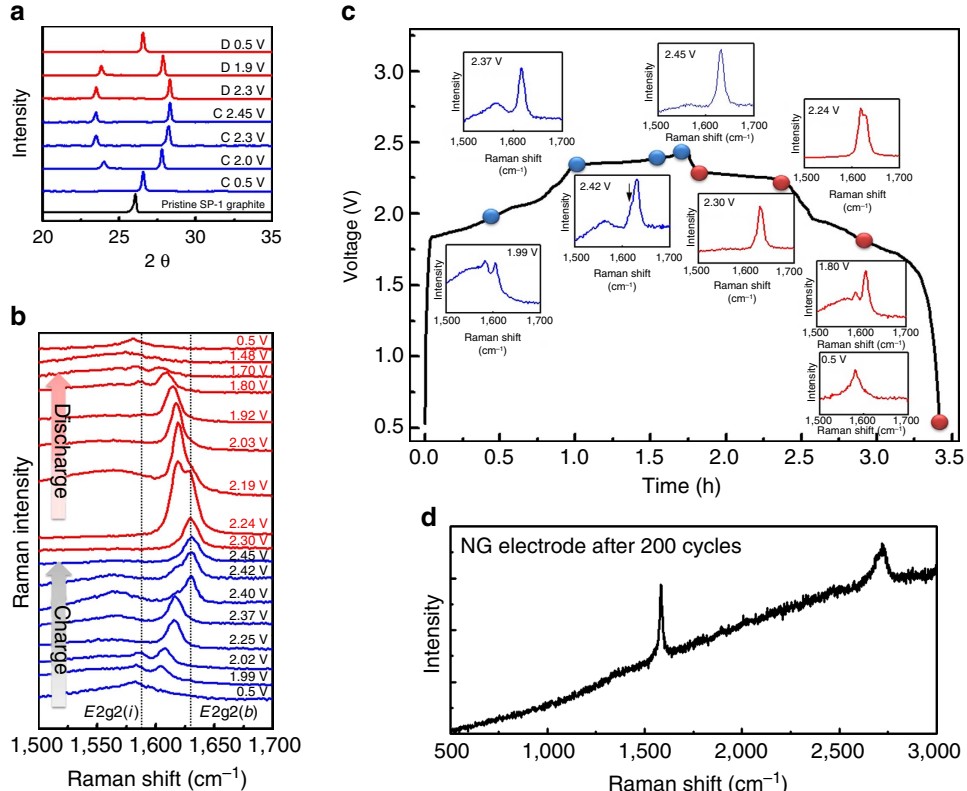

**Figure 3 | Structural and spectroscopic evolution of natural graphite.** (**a**) *Ex situ* X-ray diffraction patterns of NG in various charging and discharging states (denoted C and D in the figure, respectively) through the second cycle. (**b**) *In situ* Raman spectra recorded for the NG cathode through a charge–discharge cycle showing chloroaluminate anion intercalation/de-intercalation into graphite. (**c**) Raman spectra of NG recorded at the charging voltage of 1.99, 2.37, 2.42 and 2.45 V and discharging voltage of 2.3, 2.24, 1.8 and 0.5 V, respectively. (**d**) Raman spectrum of the NG electrode after hundreds of cycles.

X-ray diffraction peaks reverted to a single peak at $\sim 26.6°$. Importantly, we found that the full-width at half-maximum (FWHM) of the NG (002) peak when discharged to 0.5 V was the same as that of pristine NG, as well as NG charged to 0.5 V (FWHM $\sim 0.20°$ for all cases). These results suggested highly reversible structural evolution of NG with little disorder through charge/discharge cycling with intercalation/de-intercalation of chloroaluminate ions. The X-ray diffraction patterns of NG flake exhibited higher degrees of structural reversibility than pyrolytic graphite used previously[9]. In the latter case, the (002) peak did not fully recover to that of the pristine pyrolytic graphite upon discharge, suggesting residual disorder and strain in the pyrolytic graphite structure.

Figure 3b shows the *in situ* Raman spectra of chloroaluminate anion intercalation/de-intercalation reactions with NG in the cathode of an AIB during charge/discharge. Figure 3c shows the Raman spectra of NG under various charging and discharging voltages. Along the first charging plateau as the charging voltage increased to 1.99 V, a doublet peak of the $E2g2(i)$ mode (1,586 cm$^{-1}$) and the $E2g2(b)$ mode (1,607 cm$^{-1}$) appeared, which was consistent with graphite intercalation[18–20]. The doublet peaks were due to the graphite G band splitting into a lower-frequency component ($E2g2(i)$) attributed to vibrations of carbon atoms in the interior of graphite layer planes (not adjacent to intercalant layer planes), and a higher-frequency component ($E2g2(b)$), attributed to vibrations of carbon atoms in bounding graphite layers (adjacent to intercalant layer planes)[21]. Along the second charging plateau from 2.37 to 2.45 V, obvious blue shifts of the $E2g2(b)$ peak were observed, evolving into a single, dominant peak at 1,630 cm$^{-1}$ at 2.45 V (Fig. 3c). The Raman data pointed to two different intercalation processes, involving chloroaluminate anions at the two charging plateaus of $\sim 1.9$–2.37 V and 2.37–2.45 V, respectively. The first plateau in the graphite charging curve showed doublet Raman peak features from G band splitting and the higher voltage plateau showed a single, dominant blue-shifted peak.

During discharge, the opposite trends were observed when chloroaluminate anions were de-intercalated. With a decreasing voltage, the $E2g2(b)$ peak diminished, accompanied by the reappearance and growth of the $E2g2(i)$ peak to form the doublet feature (Fig. 3b,c). The original graphite spectrum was recovered when the cell was fully discharged. Moreover, Raman spectrum of the NG electrode after hundreds of cycles still showed no obvious D band, indicating that the structure of SP-1 graphite powder is maintained during the long-term cycling of chloroaluminate anions intercalation/de-intercalation (Fig. 3d).

To probe the chemical nature of the intercalated compound in our NG cathodes, *ex situ* XPS measurements were carried out (Fig. 4). To minimize trapped electrolyte, each cathode was washed with anhydrous methanol thoroughly before measurements. XPS probed the C (1s) signals of NG in the cathode during charging and discharging at the 50th cycle of an AIB. On charging as the voltage increased from 0.5 to 2.5 V, the 284.5 eV C 1s peak shifted to a higher-binding energy of 285.1 eV and upon discharge the C 1s peak fully recovered to the initial binding energy (Fig. 4a). This suggested highly reversible oxidation/reduction of carbon in NG during chloroaluminate anion intercalation/de-intercalation. Note that the C1s XPS peak in NG through charge/discharge exhibited much better defined shapes and reversibility than in the case of pyrolytic graphite[9].

To understand the effect on graphite of chloroaluminate anion intercalation at different charge and discharge stages, XAS was

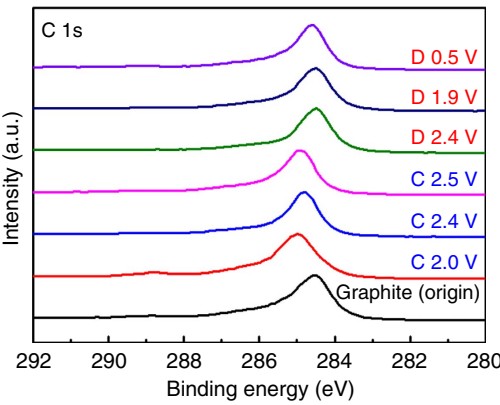

**Figure 4 | *Ex situ* X-ray photoemission spectra of graphite C 1s.** The graphite cathode in Al/NG cell was measured in various charging and discharging states (denoted C and D, respectively) through the second cycle.

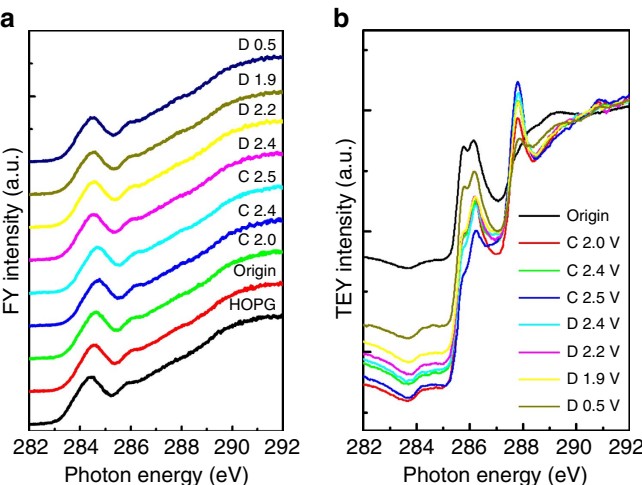

**Figure 5 | X-ray absorption spectra of graphite at C-K-edge.**
(**a**) fluorescence (fluorescence yield, FY) mode and (**b**) total electron yield (TEY) mode of natural graphite in various charging and discharging states (denoted C and D, respectively) through the second cycle. HOPG = Highly oriented pyrolytic graphite.

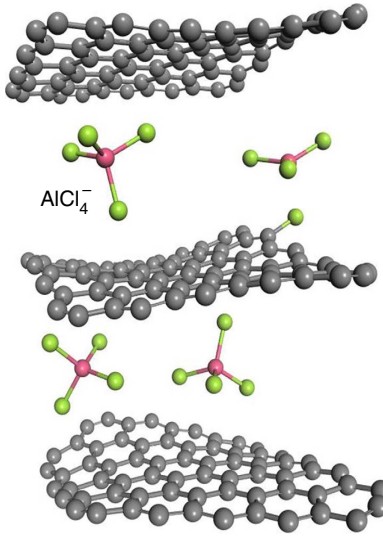

**Figure 6 | A model of chloroaluminate anions intercalating into the graphite layers.** Density functional theory (DFT) and first-principles calculations was performed to investigate the structure of four intercalated tetrahedrally structured $AlCl_4^-$ anions intercalated in the edge position of the graphite layers. The green and red spheres represent Cl and Al atoms, respectively. The carbon atoms are indicated by the grey spheres.

performed to investigate carbon's bonding environment. The carbon K-edge measured by fluorescence yield (Fig. 5a) showed that the C 1s to $\pi^*$ absorption peak at 284.5 eV shifted to higher energy (consistent with oxidation of C) on charging/anion intercalation and reversed on discharging. The reversibility indicated the stability of the bulk structures of NG flakes over repeated cycles of intercalation/de-intercalation by chloroaluminate anions. Interestingly, the carbon K-edge measured by total electron yield (TEY) reflecting the surface carbon species of NG flakes (Fig. 5b) showed that the intensity of C 1s to $\pi^*$ transition decreased appreciably after charging to various voltages. In addition, two peaks at 286 and 288 eV were found in the TEY data on charging/discharging cycles that remained after discharge back to 0.5 V. These two peaks were not observed in the fluorescence yield spectra reflecting the bulk carbon species. We attributed the peak at 286 eV to carbon in the C–N and C–H bonds of 1-ethyl-3-methylimidazolium cations in the electrolyte residue[22]. The peak observed by TEY at 288 eV—likely resulted from interactions between carbon and chloride ligand of chloroaluminate anions and C–Cl bonding on the surface or edges of NG flakes[23]. Such bonding could be a side reaction of the AIB and may be partly responsible for the non-ideal CE of the

cell, and the side reaction that occurred more readily on the edges or surface defects of NG flakes.

## Discussion
To gain a better insight into this intercalation phenomenon, we performed DFT[24–26] and first-principles calculations to investigate chloroaluminate anions intercalated into the graphite layers (Fig. 6). The *d*-spacing of the graphite model was constructed to be 8.205 Å between the three graphene layers (The *d*-spacing is 4.017 Å between two graphene layers; Supplementary Fig. 5). The simulated supercell consists of 201 carbon atoms and five $AlCl_4^-$ clusters were used with k-point, with model dimensions $20 \times 20 \times 25$ Å$^3$. The original $AlCl_4^-$ cluster size before intercalating into the graphene layer was 4.88 Å. Simulation results suggested that the intercalated tetrahedral $AlCl_4^-$ anions were distorted by the pressure of the graphite layers. The four bond angles of $AlCl_4^-$ anions in graphite layers were changed to 107.8°, 106.8°, 110.1° and 107.6° from the tetrahedral structure (109.5°) of free $AlCl_4^-$ anions. The size of distorted $AlCl_4^-$ anion was reduced to $\sim$4.79 Å, suggesting that $AlCl_4^-$ became distorted/flattened from the ideal tetrahedron structure. In spite of the behaviour of $AlCl_4^-$ intercalation, we observed that there is a possibility to form C–Cl bonding at the graphite edge, originating from a reaction between one of the chlorine atoms of the $AlCl_4^-$ anion and the carbon atom on the edge of the graphite layer. The formation of C–Cl at the surface graphite was consistent with experiment the C–K-edge spectra of TEY of XAS.

In summary, we developed an AIB using NG flake film as the cathode combined with an aluminium anode in an IL electrolyte. The specific capacity of the Al/NG battery was significantly greater than that of a similar cell made with pyrolytic graphite (that is, an increase to $\sim$110 mAh g$^{-1}$ from $\sim$66 mAh g$^{-1}$). The battery exhibited stable cycling behaviour over >6,000 charge–discharge cycles without any decay and displayed a high discharge voltage plateau. The current Al-graphite battery produces an energy density of $\sim$68.7 Wh kg$^{-1}$ (based on $\sim$110 mAh g$^{-1}$ cathode capacity and the masses of active materials in electrodes and electrolyte). Spectroscopic and

theoretical modelling has given significant insight into aluminium ion battery reactions.

## Methods

**Preparation of NG flake electrode.** First, the NG slurry was prepared by mixing SP-1 graphite powder (TED PELLA, Inc. Prod. No. 61–302, Lot# 042111) and PVDF (10%) (HSV-900) in N-methyl pyrrolidone (Alfa Aesar, 99 + %). Then, the slurry was cast on a Cu foil (Ubiq Tech. Inc. LTD, 10 μm) to form a uniform graphite film and dried at 150 °C for 2 h. The Cu foil was then etched by immersing the sample into an iron chloride (Sigma-Aldrich, 97%) solution (0.4 g ml$^{-1}$) to form a NG film. After which the NG film was rinsed with deionized water to remove the residual FeCl$_3$, and dried at 70 °C for 3 h to obtain the NG film (2 × 2.3 cm). The specific graphite loading was ∼4 mg cm$^{-2}$.

**Preparation of [EMIm]Al$_x$Cl$_y$ IL electrolytes.** A room temperature IL electrolyte was made by mixing [EMIm]Cl (Iolitec. IL0093-HP-0100, >98%) and anhydrous AlCl3 (Alfa Aesar, 99%). [EMIm]Cl was baked at 70 °C under vacuum for 16–32 h to remove residual water. [EMIm]Al$_x$Cl$_y$ IL electrolytes were prepared in an glove box under an Argon atmosphere (note: both [EMIm]Cl and AlCl$_3$ are highly hygroscopic) by mixing anhydrous AlCl3 with [EMIm]Cl, with the resulting light-yellow, transparent liquid being stirred at room temperature for 10 min. The mole ratio of AlCl3 to [EMIm]Cl was ∼1.3. The water content of the ionic liquid was determined to be 500–700 p.p.m. using a Karl Fischer coulometer (DL 39).

**Fabrication of Al/NG pouch cell.** Pouch cells were assembled in the glove box using a NG cathode (4 mg cm$^{-2}$) and an Al foil (70 mg; Ubiq Tech. Inc. LTD, 20 μm) anode, which were separated by one layer of glass fibre filter paper (Whatman, GF/D) to prevent shorting. Polymer-coated Ni bars (Ubiq Tech. Inc. LTD, 10 μm) were used as current collectors connected to NG film and Al foil by a carbon tape (TED PELLA, Inc.) for the anode and cathode, separately. The electrolyte (2 ml, prepared using AlCl$_3$/[EMIm]Cl ∼1.3 by mole) was injected and the cell was sealed by using a heating sealer (ME-200HI, Mercier Corporation). The cell was removed from the glove box and held by two glass slide with two clips for the battery performance test. The reference voltage of each charging and discharging profile is Al$^{3+}$/Al. The detailed process was obtained in Supplementary Fig. 6.

**Ex situ X-ray diffraction measurement.** For the ex situ XRD study, an Al/ NG cell (in a Pouch configuration) was charged and discharged at a constant current density of 66 mAg$^{-1}$. The reactions were stopped at different charging and discharging voltages from 0.5 to 2.45 V. After either the charging or the discharging reaction, the graphitic cathode was removed from the cell in the glove box. To avoid a reaction between the cathode and air/moisture in the ambient atmosphere, the cathode was placed onto a glass slide and then wrapped in Scotch tape. The wrapped samples were immediately removed from the glove box for ex situ X-ray diffraction measurements, which were performed on a Bruker D8-advanced instrument.

**X-ray absorption measurement.** The carbon K-edge X-ray absorption was conducted using the beamline 20A1 in the National Synchrotron Radiation Research center (NSRRC), Hsinchu, Taiwan. The 6-m spherical grating was adopted to select the photon energy during energy scanning. The NG electrodes were obtained from de-assembling pouch cells, in which the batteries have been charge–discharge for several cycles. The de-assembling procedure was conducted in a grove box to avoid to exposure in air. The electrodes were then sealed and transferred to a vacuum chamber in the experimental end-station without contact with atmosphere. Carbon K-edge XAS spectra were recorded in TEY mode.

**Raman measurement.** In situ Raman spectra were collected on a UniRAM micro-Raman spectrometer with a laser wavelength of 532 nm during cycling. The data acquisition time was normally 10 s and accumulated for 10 times. The wavelength of laser excitation source was normalized by a silicon wafer at 520 cm$^{-1}$. A thermoelectrically cooled charge-coupled device with 1,024 × 256 pixels operating at ∼60 °C was used as the detector with 1 cm$^{-1}$ resolution. The laser line was focused onto the sample using an Olympus × 50 objective, and the laser spot size was estimated to be ∼0.8–1 μm.

**Data availability.** The data that support the findings of this study are available from the corresponding authors on request.

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

## Acknowledgements

This work was funded by the Ministry of Science and Technology of Taiwan under the project number 104-2113-M-003-006-MY2, the US Department of Energy for support of the novel materials for renewable energy project (DOE DE-SC0016165), the Global Networking Talent 3.0 plan (NTUST 104DI005) from the Ministry of Education of Taiwan and supported by the Research Fund of Taishan Scholar Project of Shandong Province of China. We also thank Dr John Rick for editing our manuscript.

## Author contributions

D.-Y.W., C.-C.C., B.-J.H. and H.D. conceived the project and designed the experiments. D.-Y.W., C.-Y.W., M.-C.L., Y.-J.H. and Y.-C.C., performed material preparation, structural characterization and electrochemical measurements. C.-J.P. and B.-J.H. perform the X-ray absorption and Raman measurements. H.-L.C., H.-A.C. and C.-W.C. perform the DFT and first principle calculation. D.-Y.W., C.-Y.W., M.-C.L., C.-J.P., B.-J.H. and H.D. analysed the data. D.-Y.W. and H.D. co-wrote the paper. All authors discussed the results and commented on the manuscript.

**Additional information**

**Competing financial interests:** The authors declare no competing financial interests.

**Publisher's note**: 

