## [Peer Review File · Nature Communications]

Reviewers' comments:

Reviewer #1 (Remarks to the Author):

This research aims to improve the specific capacity of Al-ion battery by using natural graphite film with PVDF binder as the positive electrode. The performance of the resultant electrode is good in these regards. However, due to some missing information, I am still wondering to what extent this material could lead to further interests. Also, a similar electrode using graphite and carbon black with PVDF binder has been recently reported by Zhang et al at *Advanced Energy Materials* 2016, doi:10.1002/aenm.201502588. Consequently, the novelty of this work might be compromised. My overall view of the paper is that while the work is of potential interest for publication, there are at least following critical weaknesses that need to be addressed before the work can be reconsidered for publication in this journal.

1. Despite the authors highlight the improvement of specific capacity, the specific capacity (mAh/g) vs graphite loading (mg/cm²) in Fig. S2 shows that the specific capacity decreases with graphite loading. It suggests the improvement of specific capacity is only achievable when the graphite loading is low, e.g. 2-4 mg/cm² as used in this manuscript. On the other hand, according to the data of Fig. S2, an alternative plot of columbic capacity (mAh) vs graphite loading (mg/cm²) shows that a maximum columbic capacity can only be found at around 10 mg/cm². So the improvement of specific capacity has to be inevitably compromised by the decreased columbic capacity. In fact the diffusion of chloroaluminate ions to graphite in depth and electronic conductivity both might be reduced when the thickness of film is increased. This is mainly because of the dense packing of graphite flakes and blockage of PVDF. For this reason, I recommend to the authors to address this issue by including an impedance study of different thickness.

2. In Page 4 Line 139, two d-spacings were given. Presumably these values are calculated based on (0 0 2) lattice planes, but the author still need to explain the origin of this separation of peaks, especially the one at higher angle. And why there is shift between pristine graphite and 0.5 V charged graphite as shown in Fig. 3 (a), since I believe at this voltage no intercalation should occur.

3. In Page 5 Line 176, Al(2p) signal of XPS was mentioned, but cannot be found in the manuscript. It should be provided in the manuscript. In addition, Cl(2p) spectrum of XPS should also be provided to support 'the formation of C-Cl at the surface', as indicated in Page 6 Line 214.

4. The DFT calculations provide insight into the intercalation process, however, more details of the calculations need to be provided clearly. For example, what is the value of bond length(s) of Al-Cl and geometric size of AlCl₄⁻? Is this geometric size compatible to interlayer spacing of graphite measured by XRD? How the interlayer spacing of graphene is determined, and is it flexible or rigid while doing DFT calculation?

5. The author should explain the reason why the film was cast on the copper foil followed by dissolution rather than directly cast on the nickel bar. And in Page 3 Line 90, it was mentioned that the film "was bonded to Ni bar current collectors by sandwiching conducting carbon tapes between the NG film and Ni bars". But in Fig. 1 (c), only two layers can be identified. The three layers should be clearly labelled.

6. It is expected that a halogenated ionic electrolyte could potentially undergo toxic gas formation. Is there a provision for gas entrapment during the formation cycle? If so, please describe. If not, explain why such a gas entrapment provision has not been provided.

Reviewer #2 (Remarks to the Author):

This manuscript reports a natural graphite cathode for AIBs, which has been firstly reported in *Nature* 520, 324-328. The originality regarding a new electrode material is poor. The manuscript is well written and organized. An distinct enhancement in the electrochemical properties such as reversible capacity and rate capability of the natural graphite cathode was achieved. The advance in this manuscript could be a new structural design of natural graphite cathode, and however, this

is only a routine method for electrode preparation using slurry (including NMP, PVDF and active material) coating in rechargeable batteries. The publication of this work in the top journal Nature Communications is not recommended. The detailed comments are as follows,

1. The routine slurry coating using PVDF as binder can bond the active powders with each other and also the coating layer with current collector, providing good electrical contact and mechanical performance during cycling. Therefore, it is reasonable that the nature graphite electrode prepared by slurry coating method showed a better electrochemical performance than the dense electrode prepared by only cold pressing without any binder. Actually, this rule is applicable to most electrode materials. Although this method was not used in the previous NATURE work, it does not mean herein the slurry coating could be regarded as a technical advance.

2. Besides the bonding effect of PVDF, the thickness and the mass loading are also important factors that affect the mechanical performance of the electrodes during cycling.

3. Herein the nature graphite electrode is a loose and powder electrode, which has a much higher specific surface area than the dense pyrolytic graphite foil electrode (Nature 520, 324-328.). In general, this facilitates the ionic diffusion and charge transfer at the cathode side. Consequently, a better charge/discharge reversibility and higher capacity and rate capability could be achieved.

4. The charge and discharge curves as well as the electrode characterizations at the first cycle are important in rechargeable batteries.

5. other comments

** A suggestion: the electrode film formed by the slurry coating could be readily separated from smooth substrate such as glass sheet.

** Page 2, line 43, error "804 Ah cm⁻³"

** Page 4, line 142, not clear.

Reviewer #3 (Remarks to the Author):

On the basis of the authors' former work, here it reported another improvement on the positive electrode. It is clear that the results are very interesting. However, there are some questions remained:

1. In this battery system, the cost is a crucial problem though it was claimed to be cheap. The main reason is the use of the ionic liquid electrolytes, which are very difficult to treat since here the mixture of [EMIm]Cl and aluminum chloride is used, which is sensitive to water. As a result, it should be compared with some pioneering work such as zinc/aluminum Ion Battery, which was recently published in ACS Applied Materials & Interfaces.

2. There are some mismatching saying. In the case of battery, the capacity of the NG is about 110 mAh/g. As a result, the current for 1C should be 110 mA/g instead of the claimed 99 mA/g. All the sayings about C rate should be changed.

3. In the practical lithium ion batteries, the Coulomb efficiency for every should be at least 99.9% so that the cycling can be above 2000. In this case, the negative electrode is too overdose. The authors should comment the practical cycling if the negative electrode is only about 5% overdose.

Reply to Referees.

We would like to thank the reviewers for their insightful comments. The following is our point-to-point response to the concerns raised.

Reviewer #1 (Remarks to the Author):

Comment 1. I recommend to the authors to address this issue by including an impedance study of different thickness.

Reply to the reviewer:

We thank the reviewer for this valuable comment. We have added the data from additional experiments including the EIS data of the batteries fabricated by loading different amounts of graphite under charging at 2.4 V in the supporting information. Diffusion can create an impedance which is known as the Warburg impedance. This impedance is related to the frequency (ω) of the potential perturbation. Equation 1 is for the "infinite" Warburg impedance. With mobile anions, diffusion flux implies a Warburg-like impedance; this impedance is a $\omega^{-1/2}$ function.

$$Z = \sigma(\omega)^{-\frac{1}{2}}(1 - j) \quad (\text{eq. 1})$$

On a Nyquist plot the infinite Warburg impedance appears as a diagonal line with a slope of 0.5. On a Bode plot, the Warburg impedance shows a phase shift of 45°. In the above equation, σ is the Warburg coefficient defined as:

$$\sigma = \frac{RT}{n^2 F^2 A \sqrt{2C} \sqrt{D}} \quad (\text{eq. 2})$$

ω is radial frequency, D is diffusion coefficient of anions, A is surface area of the electrode, n is the number of electrons transferred and C is the bulk concentration of the diffusing species (moles/cm³).

In Figure R1 (b), we found that the slope of the relationship between Z' and $\omega^{-1/2}$ in the battery with high a loading amount is higher than that with a low loading amount. According to eq.2, this result indicated that diffusion coefficient anions in the battery with a high loading is less than that with a low loading. The result could be attributed to that the diffusion of chloroaluminate ions to graphite may be reduced when the thickness of film is increased, resulting in a reduced specific battery capacity with higher graphite loadings at certain C-rate.

Figure R1. (a) EIS plots of the batteries with various graphite loading. (b) The relationship between Z' and $\omega^{-1/2}$ of the related EIS plots.

Revision made:

We have added the above EIS plots of the batteries with various graphite loadings, while the relationship between Z' and $\omega^{-1/2}$ are shown in EIS plots, see Figure S6 in the supporting information (Page S6).

Comment 2. In Page 4 Line 139, two d-spacings were given. Presumably these values are calculated based on (002) lattice planes, but the author still need to explain the origin of this separation of peaks, especially the one at higher angle. And why there is shift between pristine graphite and 0.5 V charged graphite as shown in Fig. 3 (a), since I believe at this voltage no intercalation should occur.

Reply to the reviewer:

Because the XRD data were collected from the battery after 50 cycles, a slight peak shift of 002 lattice planes of graphite with charging or discharging at 0.5V could be caused by few anions remaining in the graphite layer (incomplete deintercalation) at 0.5V.

The periodic repeat distance (l_c), the intercalant gallery height (d_i) and the gallery expansion (Δd) can be calculated using the following formula:

$$l_c = d_i + 3.35\text{\AA} \times (n - 1) = \Delta d + 3.35\text{\AA} \times n = l \times d_{obs}$$

During the charging/anion-intercalation process, the (002) peak completely vanished and gradually split into two new peaks signaling chloroaluminate anion intercalation into NG. The two peaks correspond to expansion and compression of the d-spacing respectively through various regions of graphite during intercalation. The intensity pattern is commonly found for a stage n graphite intercalation compound (GIC), where the most dominant peak is the (00n+1) and the second most dominant peak is the (00n+2). Based on our experimental data, by increasing the charging state to the fully charged state, the distance between the (00n+1) and (00n+2) peaks gradually increased, as more AlCl_4^- anions intercalated. The d spacing values of (00n+1) and (00n+2) peaks (that is, $d_{(n+1)}$ and $d_{(n+2)}$, respectively) were calculated from XRD data (Figure 3a). The most dominant stage phase of the observed GIC can be assigned by determining the ratio of the $d_{(n+2)}/d_{(n+1)}$ peak position which correlated these to the ratios of stage pure GICs. Therefore, a stage 4 graphite intercalated by AlCl_4^- anions and $d_{obs} \sim 3.8 \text{ \AA}$ (peak at $2\theta \sim 23.2^\circ$) was determined by analysis of the peak separation.

Comment 3. In Page 5 Line 176, Al(2p) signal of XPS was mentioned, but cannot be found in the manuscript. It should be provided in the manuscript. In addition, Cl(2p) spectrum of XPS should also be provided to support 'the formation of C-Cl at the surface', as indicated in Page 6 Line 214.

Reply to the reviewer:

Because we found no more information related to anion intercalation in the graphite layer from the Al(2p) spectra of XPS, the Al (2p) spectra have been removed. Also an XAS spectra of Cl(2p) showed that there are no obvious differences between the samples at different stages (see below). This finding may be attributed to the Cl signal being a sum of AlCl_4^- or Al_2Cl_7^- in the electrolyte and in the intercalated layers, leading to difficulties in distinguishing the differences between ions in electrolyte and in graphite. Future work is needed using spatially resolved X-ray tools (*e.g.*, STXM).

Figure R2. X-ray absorption spectra of graphite and AlCl_4^- electrolyte at Cl K-edge.

Comment 4. The DFT calculations provide insight into the intercalation process, however, more details of the calculations need to be provided clearly. For example, what is the value of bond length(s) of Al-Cl and geometric size of AlCl_4^- ? Is this geometric size compatible to interlayer spacing of graphite measured by XRD? How the interlayer spacing of graphene is determined, and is it flexible or rigid while doing DFT calculation?

Reply to the reviewer:

In the DFT section, the Al-Cl bond length lengths are listed in Table S2. AlCl_4^- clusters, with tetrahedron and planar quadrangle geometries, were separately inserted into

the relaxed graphite, as shown in Fig. 6. The AlCl_4^- cluster size, before intercalation into the graphene layer, was 4.88 Å. We modeled graphite as 3-layer graphene. The d -spacing of the graphite was measured to be 8.205 Å between the three graphene layers, (note: the d -spacing is 4.017 Å between two graphene layers), which is slightly larger than the d spacing of bulk graphite (3.4 Å). Our calculated result is based on d -spacing that allows for two graphene layers with Van der Waals radii. The simulated supercell consists of 201 carbon atoms and five AlCl_4^- clusters which were used with k-Point, and model dimensions $20 \times 20 \times 25 \text{ Å}^3$. When compared to the AlCl_4^- cluster in the vacuum space, the intercalated AlCl_4^- cluster showed clear distortions in both the calculated Al-Cl bond lengths and bond angles. The distortion directly results from the Van der Waals interactions between the graphene layers, which leads to flattening of the AlCl_4^- tetrahedron in response to pressure from the c -axis direction. The interlayer graphene spacing was substantially enlarged (6.231 Å) after AlCl_4^- intercalation (see figure below), with the geometric size (4.79 Å) of the intercalated anion. The spacing in the DFT calculation is slightly smaller than the interlayer spacing of graphite as measured by XRD (the gallery height $d_i \sim 8.8 \text{ Å}$, see comment 2). The reason could be attributed to the scale of our DFT model is too small in comparison with real size of graphite flake (average size $\sim 100 \mu\text{m}$).

Figure R3. The DFT models of (a) the graphite with three layers and (b) the graphite with intercalatants.

Revision made:

We have added the above detailed simulation information and the anion-size/graphite d spacing analysis, including two figures (Figure S5) in the **Page S1-S2 and S6** of supporting information.

Comment 5. The author should explain the reason why the film was cast on the copper foil followed by dissolution rather than directly cast on the nickel bar. And in Page 3 Line 90, it was mentioned that the film "was bonded to Ni bar current collectors by sandwiching conducting carbon tapes between the NG film and Ni bars". But in Fig. 1 (c), only two layers can be identified. The three layers should be clearly labelled.

Reply to the reviewer:

Casting graphite slurry on Ni substrate was done in our preliminary experiment. The Ni foil was used as substrate for slurry instead of Ni bar since the area of Ni tab is

too small to be cast. Our results showed that the capacity is similar to the graphite electrode on carbon tape. However, poor cyclability was observed – possibly due to side reactions related to Ni and the electrolyte.^{ref1} The capacity drops seriously after around 150 cycles. This might be due to the corrosion of Ni surface in electrolyte solution during cycling and graphite particles which bind to the Ni surface resulting in the loss of electrical contact and hence degrading the capacity. Because the surface area of the Ni tab (0.3cm*3cm) is much smaller than Ni foil ((2.3cm*2cm)), the corrosion extent of Ni tab will be lower than that of Ni foil, resulting in the better stability of our graphite cathode electrode with using Ni tab.

Ref1: Nakaya, K et.al Journal of The Electrochemical Society, 162 (1) D42-D48 (2015).

Figure R4. Performance of the aluminum ion battery which is using Ni foil as a current collector in the cathode electrode. (a) Galvanostatic charge and discharge curves of an Al/NG pouch cell at a current density of 99 mA/g. (b) Long-term stability test of an Al/NG cell at 99 mA/g. The capacity of Al/NG battery was recorded between charging and discharging voltage of 0.5V and 2.45 V. All graphite loading mass of batteries is $\sim 6 \text{ mg/cm}^2$

Revision made:

We have added above figure, some sentences and references to state side reaction of Ni corrosion in the ionic electrolyte (see in the supporting information, **Page S7**).

Comment 6. It is expected that a halogenated ionic electrolyte could potentially undergo toxic gas formation. Is there a provision for gas entrapment during the formation cycle? If so, please describe. If not, explain why such a gas entrapment provision has not been provided.

Reply to the reviewer:

GC-MS spectroscopy of gaseous samples withdrawn from the Al/GF cells after 30 cycles was run. H₂ gas was obviously found when the battery was charged to 4V. But no Cl₂ gas evolution was found. When compared with the background data some peaks appear that originate from contaminants in the GC/MS chamber. (see Figure R5)

Figure R5. Mass spectrum of Al ion battery with charging to 4V.

Revision made:

We have added the above GC-MS data (see Figure S8) and a related explanation in the supporting information (**Page S8**).

Reviewer #2 (Remarks to the Author):

Comment 1. The routine slurry coating using PVDF as binder can bond the active powders with each other and also the coating layer with current collector, providing good electrical contact and mechanical performance during cycling. Therefore, it is reasonable that the nature graphite electrode prepared by slurry coating method showed a better electrochemical performance than the dense electrode prepared by only cold pressing without any binder. Actually, this rule is applicable to most electrode materials. Although this method was not used in the previous NATURE work, it does not mean herein the slurry coating could be regarded as a technical advance.

Comment 2. Besides the bonding effect of PVDF, the thickness and the mass loading are also important factors that affect the mechanical performance of the electrodes during cycling.

Reply to the reviewer:

We thank the reviewer for his comments. This work has made major advances in Al ion battery when compared to the previous 'Nature work' for the reasons below (note: the PVDF binder is not one of them).

- In this study, our breakthrough was the development of a new rechargeable AIB with a high capacity (over 100 mAh/g) by using sp-1 nature graphite film as the AIB cathodic electrode. By using sp-1 graphite, our AIB cell exhibited clear discharge voltage plateaus in the ranges 2.25 to 2.0 V and 1.9 to 1.5 V, while the graphite cathode showed a much improved specific capacity over pyrolytic graphite up to ~110 mAh/g and ~ 98% Coulombic efficiency at a current density of ~ 100 mA/g. In comparison with our previous work, which published in *Nature* 520, 325-328 (2015), AIB used pyrolytic graphite papers

as cathode and yielded a relative low capacity, *i.e.* of ~ 66 mAh/g. There should be ample room for investigating graphite materials with high chloroaluminate anion intercalation capacity with minimum side reactions.

- Further, it is important to investigate and understand the chloroaluminate anion-graphite intercalation reactions in relation to the electrochemical voltage plateaus, which could hold a key to future battery development. Therefore, in this work, facilitated by this high performance battery, the detailed characterization of reversible structural evolutions of NG film during charging/discharging process were performed by *ex situ* XRD, XAS XPS and *in situ* Raman measurements. From these results, we found some abnormal transformation trends of graphite and side effects during anion intercalation and deintercalation processes. We also proposed a new mechanism for the intercalation behavior of chloroaluminate anions in graphite layers - by using DFT and a first principle simulation in conjunction with XAS and Raman Results. Here are the new scientific points of view that appear in our current manuscript. I think that the reviewer has perhaps misunderstood some of the points in our manuscript. We agree that the use of PVDF as binder to fabricate the graphite slurry is a common process used in lithium ion battery manufacture.

Compared to our previous work published in Nature, this is the first time to report that what kind of graphite structure will achieve high capacity during anion intercalation and to propose new structural information and side reaction for AlCl_4^- anions intercalated in the graphite layers. In the supporting information, different types of graphite with higher defect structures including MCMB and KS6 in comparison with sp-1 graphite were also used as cathode electrodes and exhibited less capacity and Coulombic Efficiency. We found that the performance of the Al ion battery is strongly depended on the high crystallinity and low defect structure of graphite. Therefore, we believe that our creation of an Al ion battery using sp-1 graphite is a step-forward towards increasing the energy density of Al ion batteries. Moreover, the calculations were performed to explain the anion intercalation mechanism and the structural transformation that occurs in the graphite layers. Overall, we have proposed a good approach to enhancing the anion specific capacity of the graphite using well characterized materials the usage of which is supported by a clearly presented scientific rationale. Several points related to intensive scientific information are as follows:

1. Battery performance achievement: The specific capacity and Coulombic efficiency of Al/NG cell was up to ~ 110 mAh/g and $\sim 98\%$
2. In situ Raman measurements: Different intercalation processes for the chloroaluminate anions were identified at the two charging plateaus.
3. The battery performance strongly depended on the crystallinity and defect of graphite. Our findings are different to those for carbon materials used in Li ion batteries. Two synthetic graphite materials, KS6 and MCMB widely used as Li ion battery anode materials, were found to give much lower capacity than that of NG without clear discharge voltage plateaus (see Fig S1). This was attributed to that KS6 and MCMB materials exhibiting lower crystallinity and higher defect density than natural graphite based on Raman and XRD measurements (Fig. S1). The results indicated that graphite materials with high crystallinity and low defect density are

needed for high performance Al ion batteries.

4. X-ray absorption spectroscopy: possible formation of C-Cl bonding on the surface or edges of graphite flakes.
5. DFT simulation result: Intercalated AlCl_4^- anions with tetrahedral structure were distorted by the pressure of graphite layers.
6. Theoretical result by DFT: a C-Cl bonding at the edge was formed when one of the chlorine atoms of AlCl_4^- anion moved to the edge position of the carbon layer.

Comment 3. Herein the nature graphite electrode is a loose and powder electrode, which has a much higher specific surface area than the dense pyrolytic graphite foil electrode (Nature 520, 324-328.). In general, this facilitates the ionic diffusion and charge transfer at the cathode side. Consequently, a better charge/discharge reversibility and higher capacity and rate capability could be achieved.

Reply to the reviewer:

In fact, we found that Al ion batteries made using different kinds of graphite exhibit different performance characteristics (see supporting information Figure S1). In our manuscript, we found that the battery performance strongly depends on the crystallinity and defect structure of graphite and is not just related to loose and powder electrodes. Graphite with a high surface area will cause more side reactions during charging and discharging, resulting in less battery performance (low specific capacity and Coulombic efficiency). This conclusion is a very important scientific point that paves the way to further improving battery performance by using high crystallinity and low defect graphite.

Revision made:

We have added a sentence to explain the differences between natural graphite and KS6 or MCMB materials and related battery performance in the revised manuscript (see **Page 4**) and supporting information (see **Page S2**, Figure S1).

Comment 4. The charge and discharge curves as well as the electrode characterizations at the first cycle are important in rechargeable batteries.

Reply to the reviewer:

In the Al ion battery, there is no SEI formation in the first cycle which is different from Li ion battery. In the first cycle of the Al ion battery, the structure of sp² graphite is expanded by anion intercalation. We added *in situ* Raman spectra of graphite at first cycle in the supporting information. The splitting and shifting behavior of graphite G band at first cycle is slightly different from that at second cycle (Figure R6 (a-b)), the result showed that the transition between 1620 and 1630 at charging to 2.4V is not obvious in the first cycle. This could be attributed to the expansion of the graphite layer structure, resulting in lower Coulombic Efficiency, see Figure R6 (c-d).

Figure R6. *In situ* Raman spectra recorded for the NG cathode at first (a) charge and (b) discharge cycle showing chloroaluminate anion intercalation/de-intercalation into graphite. (c) Different galvanostatic charge and discharge cycles of an Al/NG pouch cell were recorded at a current density of 99 mA/g. (d) The related cycle performance of Al/NG cell.

Comment 5. other comments

** A suggestion: the electrode film formed by the slurry coating could be readily separated from smooth substrate such as glass sheet.

** Page 2, line 43, error "804 Ah cm⁻³"

** Page 4, line 142, not clear.

Revisions made:

Thanks for reviewer's comment. These typos have been corrected.

Reviewer #3 (Remarks to the Author):

Comment 1. In this battery system, the cost is a crucial problem though it was claimed to be cheap. The main reason is the use of the ionic liquid electrolytes, which are very difficult to treat since here the mixture of [EMIm]Cl and aluminum chloride is used, which is sensitive to water. As a result, it should be compared with some pioneering work such as zinc/aluminum Ion Battery, which was recently published in ACS Applied Materials & Interfaces.

Reply to the reviewer:

Thanks for reviewer's comment. We have added the description, comparison and ref. about Zinc/Al ion battery in our manuscript.

Revision made:

We have added the description, comparison and ref. about Zinc/Al ion battery in the revised manuscript. A sentence "graphite materials have been used in aqueous rechargeable zinc/aluminum ion batteries that use zinc as the negative electrode and ultrathin graphite nanosheets as the positive electrode in an aqueous Al₂(SO₄)₃/Zn(CHCOO)₂ electrolyte.¹⁶" was added into Page 2 of the revised manuscript. The new ref was also added (see Ref 16, page 10).

Comment 2. There are some mismatching saying. In the case of battery, the capacity of the NG is about 110 mAh/g. As a result, the current for 1C should be 110 mA/g instead of the claimed 99 mA/g. All the sayings about C rate should be changed.

Reply to the reviewer:

We agree with the reviewer's valuable suggestion. The C rate in our manuscript has been changed to "current for 1C is 110 mA/g". the C-rate was recalculated and corrected in revised manuscript.

Revision made:

We have changed all c-rate in our revised manuscript.

1. 99mA/g (1C) is changed to 99mA/g (0.9C), (**page 1**, abstract part)
2. 660mA/g (6.6C) is changed to 660mA/g (6C), (**page 1**, abstract part, **page 4** line 7 and line 10)
3. (2C) 198 mA/g is changed to (1.8C) 198 mA/g, (**page 4** line 12)

Comment 3. In the practical lithium ion batteries, the Coulomb efficiency for every should be at least 99.9% so that the cycling can be above 2000. In this case, the negative electrode is too overdose. The authors should comment the practical cycling if the negative electrode is only about 5% overdose.

Reply to the reviewer:

Thanks for reviewer's comment. In the Li ion battery, the reason the anode electrode was overdosed with 3-15% capacity was to compensate for SEI formation, which results in 'consumption' of Li ions, at the anode during the 1st cycle. Therefore, overdosing helps to avoid lithium dendrite formation in the following cycles and thereby improves issues related to safety. However, in Al ion battery the anode reaction is in principle a perfect Al deposition and dissolution with no sign of SEI formation being found. Therefore, we expect that the extent of overdose, even at 5%, will not exert a critical effect on the cycling performance of the Al ion battery.

Reviewers' comments:

Reviewer #1 (Remarks to the Author):

The authors of this manuscript have demonstrated a rechargeable AIB using a film of natural graphite flakes with a polyvinylidene fluoride binder as the cathode of a rechargeable Al-ion battery. Although, the performance of the resultant electrode is good in these regards. However, the main results of such battery have been reported by their previous work on Journal of Nature, and also few groups disclosed the similar battery. In this case, the work is not enough for publishing in NC. Some of specific comments are as following.

1. What was the open circuit (no-load) voltage? How stable was it over time? And what is the reference voltage in this system (Al^{3+}/Al or what)?
2. The electrochemical analysis lack some important information. CV is the most basic but provide essential information.
3. The manuscript is missing quite a bit of the materials and methods, and so I recommend that the authors should add more experimental details. For example, is the electrolyte used as is, or was it electrochemically cleaned first? How was the pouch cell assembled?
4. In this manuscript, the authors adopted Ni bar as current collector, and we think that Ni current collector begins to dissolve at around 1.3 V in this ionic liquid electrolyte system, so why did the authors choose the Ni as current collector? The authors should explain the reason in detail.
5. In Fig. R5, It was mentioned that H_2 gas was obviously found when the battery was charged to 4 V. Why the battery was charged to 4 V? And the authors should explain or mark the characteristic peak of H_2 .
6. The authors should pay attention to the written form in the references (such as: case-sensitive, superscripts and subscripts and so on).

Reviewer #2 (Remarks to the Author):

The authors have carefully improved the manuscript by addressing the reviewers' comments. The original scientific advances have been shown more clear. I would like to accept publication of this work in Nature Communications.

Reviewer #3 (Remarks to the Author):

It is recommended to publish in this journal.

Reply to Referees.

The following is our point-to-point response and revisions made to fully address the concerns raised by reviewer 1.

Reviewer #1 (Remarks to the Author):

The authors of this manuscript have demonstrated a rechargeable AIB using a film of natural graphite flakes with a polyvinylidene fluoride binder as the cathode of a rechargeable Al-ion battery. Although, the performance of the resultant electrode is good in these regards. However, the main results of such battery have been reported by their previous work on Journal of Nature, and also few groups disclosed the similar battery. In this case, the work is not enough for publishing in NC. Some of specific comments are as following.

Reply to reviewer 1

In our previous work published in Nature and other report (*Chem. Commun.*, 2015, 51, 11892–11895,), Al ion battery used pyrolytic graphite papers (Nature) or graphite paper (*Chem. Commun.*, 2015, 51, 11892–11895,) as cathode and yielded a relative low capacity, i.e. of ~ 70 mAh/g. There should be ample room for investigating graphite materials with high chloroaluminate anion intercalation capacity with minimum side reactions. Therefore, this is the first time to report that what kind of graphite structure will achieve high capacity during anion intercalation and to propose new structural information and side reaction for AlCl_4^- anions intercalated in the graphite layers. In this work, we have proposed a good approach to enhancing the anion specific capacity of the graphite using well characterized materials the usage of which is supported by a clearly presented scientific rationale. Several points related to intensive scientific information are as follows:

1. Battery performance achievement: The specific capacity and Coulombic efficiency of Al/NG cell was up to ~ 110 mAh/g and ~98%
2. In situ Raman measurements: Different intercalation processes for the chloroaluminate anions were identified at the two charging plateaus.
3. The battery performance strongly depended on the crystallinity and defect of graphite. Our findings are different to those for carbon materials used in Li ion batteries. Two synthetic graphite materials, KS6 and MCMB widely used as Li ion battery anode materials, were found to give much lower capacity than that of NG without clear discharge voltage plateaus (see Fig S1). This was attributed to that KS6 and MCMB materials exhibiting lower crystallinity and higher defect density than natural graphite based on Raman and XRD measurements (Fig. S1). The results indicated that graphite materials with high crystallinity and low defect density are needed for high performance Al ion batteries.
4. X-ray absorption spectroscopy: possible formation of C-Cl bonding on the surface or edges of graphite flakes.
5. DFT simulation result: Intercalated AlCl_4^- anions with tetrahedral structure were distorted by the pressure of graphite layers.
6. Theoretical result by DFT: a C-Cl bonding at the edge was formed when one of the chlorine atoms of AlCl_4^- anion moved to the edge position of the carbon layer.

The following is our point-to-point response and revisions made to fully address the concerns raised by reviewer 1.

Comment 1. What was the open circuit (no-load) voltage? How stable was it over time? And what is the reference voltage in this system (Al^{3+}/Al or what)?

Reply to the reviewer:

Thanks for reviewer's suggestion. The OCV of Al/GF cell is around 1.0V and the cell is quite stable for three months stability test (Figure 2c). The reference voltage of each charging and discharging profile is Al^{3+}/Al which was added into the experimental section (marked in red).

Revision made:

We have added the sentence "The reference voltage of each charging and discharging profile is Al^{3+}/Al " into the revised manuscript. (Page 7)

Comment 2. The electrochemical analysis lack some important information. CV is the most basic but provide essential information.

Reply to the reviewer:

Thanks for reviewer's valuable suggestion. The CV scan has been added in the revised supporting information.

Figure R1. Cyclic voltammetry curves of Al foil and NG at a scan rate of 0.1 mV/s against an Al reference electrode.

Revision made:

The cyclic voltammetry curves of Al foil and NG were added into the revised supporting information (see Figure S9) (Page S8).

Comment 3. The manuscript is missing quite a bit of the materials and methods, and so I recommend that the authors should add more experimental details. For example, is the electrolyte used as is, or was it electrochemically cleaned first? How was the pouch cell assembled?

Reply to the reviewer:

Thanks for reviewer's suggestion. Our raw materials of the electrolyte was only baked at 70 °C under vacuum for 16–32 h to remove residual water and then the electrolyte was prepared in the glove box without electrochemical cleaning. More experimental details have been added into experimental section and supporting information (marked in red).

Figure R2. The detailed fabrication of Al/NG cell assembled in the pouch cell. (a) Pouch cells were assembled by using a NG cathode (4 mg/cm^2) and an Al foil (70 mg) anode, which were separated by one layer of glass fiber filter paper to prevent shorting. Polymer coated Ni bars were used as current collectors connected to NG film and Al foil by a carbon tape for the anode and cathode, separately. (b) The two sides of resulting pouch cell were sealed by a heating sealer for further adding electrolyte easily in the glove box. (c) The resulting pouch cell was delivered into the glove box. Then, the electrolyte (2mL, prepared using $\text{AlCl}_3/[\text{EMIm}]\text{Cl}$ ~1.3 by mole) was injected and the cell was sealed by a heating sealer. (d) Finally, the cell was prepared and removed from the glove box for further performance test.

Revision made:

The more experimental details (Figure S10) have been added into the revised supporting information. (Page S9)

Comment 4. In this manuscript, the authors adopted Ni bar as current collector, and we think that Ni current collector begins to dissolve at around 1.3 V in this ionic liquid electrolyte system, so why did the authors choose the Ni as current collector? The authors should explain the reason in detail.

Reply to the reviewer:

Thanks for reviewer's comment. To make sure that Ni bar could have corrosive action in the range of charging and discharging voltage, a CV scan using Ni bar as working electrode, Al as ref and counter electrode in the ionic liquid electrolyte was obtained. In figure R3, We found a much lower current in the voltage range of 0.5V to 2.45V in comparison with that of graphite, indicating no severe corrosive behavior on Ni bar. However, we are still trying to find a better alternative current collector such as carbon coated metal substrates for further preventing the corrosion problem.

Figure R3. (a) Cyclic voltammetry curves of Al foil, NG and Ni bar at a scan rate of 0.1 mV/s against an Al reference electrode. (b) Enlarged cyclic voltammetry curves of Ni bar at a scan rate of 1 mV/s against an Al reference electrode.

Comment 5. In Fig. R5, It was mentioned that H₂ gas was obviously found when the battery was charged to 4 V. Why the battery was charged to 4 V? And the authors should explain or mark the characteristic peak of H₂.

Reply to the reviewer:

Thanks for reviewer's comment. The purpose of the battery charged to higher voltage (4V) was to make sure no generation of Cl₂ gas in the ionic electrolyte consisting of AlCl₄⁻ and EMI⁺. The H₂ peak has been marked in the revised supporting information.

Revision made:

We have changed Figure S8 including the H₂ peak marked in GC-MS spectrum (**Page S8**)

Comment 6. The authors should pay attention to the written form in the references (such as: case-sensitive, superscripts and subscripts and so on).

Revision made:

Thanks for reviewer's suggestion. The format of the references in the revised manuscript has been corrected. (**Page 8**)

Reviewer #2 (Remarks to the Author):

The authors have carefully improved the manuscript by addressing the reviewers' comments. The original scientific advances have been shown more clear. I would like to accept publication of this work in Nature Communications.

Reviewer #3 (Remarks to the Author):

It is recommended to publish in this journal

Reply to Reviewer 2 and 3

We would like to thank the reviewers for their insightful comments and strong positive recommendations by reviewer 2 and 3.

REVIEWERS' COMMENTS:

Reviewer #1 (Remarks to the Author):

Many thanks for having the chance to review Al-Graphite again. Partly, the work is interesting. However, such kind battery has been reported by Dai's and Jiao's group in 2015, and Jiao's group patented in 2014. Actually, the graphite cathode for Al battery reported few years earlier by one of indian research group. At current stage, the energy density is a big problem for the whole cell. It is possible for Al-C battery industry application only with getting higher energy battery than that of lead-acid battery. In reviewer's group, we reported Ah-lever battery (Carbon, 2016,109,276). However, we are still on the way to improve the energy density. If the author can show the result with more than 40Wh/kg, the work is publishable. Otherwise, the scientist should work on solving the main issue of Al-ion battery. Therefore, the reviewer is not going to recommend the paper is published on NC.

Reply to Referees.

The following is our response made to fully address the concerns raised by reviewer.

Reviewer #1 (Remarks to the Author):

Many thanks for having the chance to review Al-Graphite again. Partly, the work is interesting. However, such kind battery has been reported by Dai's and Jiao's group in 2015, and Jiao's group patented in 2014. Actually, the graphite cathode for Al battery reported few years earlier by one of Indian research group. At current stage, the energy density is a big problem for the whole cell. It is possible for Al-C battery industry application only with getting higher energy battery than that of lead-acid battery. In reviewer's group, we reported Ah-lever battery (Carbon, 2016,109,276). However, we are still on the way to improve the energy density. If the author can show the result with more than 40Wh/kg, the work is publishable. Otherwise, the scientist should work on solving the main issue of Al-ion battery. Therefore, the reviewer is not going to recommend the paper is published on NC.

Reply:

We thank the reviewer's comments. We have done a survey of Al-ion batteries related patents and literature. Below, we'd like to clarify the issues brought up by the reviewer.

1. Timeline for Stanford patent and Jiao's patent.

- (1) Our group filed a patent on Al anode-graphite cathode battery using EMIC/ AlCl_3 ionic liquid electrolyte through Stanford (docket number S14-027):
 - 1st US provisional filed on Feb 28, 2014
 - 2nd US provisional filed on Nov 6, 2014
 - PCT application filed on Feb 27, 2015
- (2) A Chinese patent application on Al ion battery using EMIC/ AlCl_3 electrolyte, Al anode and graphite cathode was filed by the group of Dr. Shuqiang Jiao in the State Key Laboratory of Advanced Metallurgy, University of Science and Technology Beijing, Beijing 100083, P. R. China. The timeline for their patent is as follows.
 - Filed in China on Aug 22, 2014 (no earlier priority date, No. 201410419495.1)
 - Published on Dec 24, 2014 (CN 104241596 A)

So in terms of patent: the fact is our US provisional patent filing above was 6 months ahead of Jiao's Chinese patent filing.

2. Timeline for our Al ion battery paper vs. Jiao's paper.

We started our Al-ion battery work at Stanford on June 1st, 2013 through a collaboration with Industrial Technology Research Institute (ITRI) in Taiwan. A few months later, we found that when graphite particles were added as a conducting additive to the cathode slurry material (such as sulfur and some oxides) or when certain carbon fiber paper was used as current collector, we observed appreciable charging capacity of the cathode in the $\text{AlCl}_3/\text{EMImCl}$ ionic liquid electrolyte. This led us to focusing on graphite and the discovery that high quality graphite is a key cathode material for the development of Al ion battery.

We first submitted our Al-ion battery manuscript to Science in 2014 with a manuscript number 1254249 entitled "A Rechargeable Aluminum Battery with High Voltage, Stability and Bendability". The manuscript was peer-reviewed and rejected on May 16, 2014. We took the reviewers' advices, performed additional experiments, improved the manuscript and submitted the paper to Nature. The paper was published in Nature on 16 April 2015 (Nature 520, 324-328).

Jiao et al. published their Al-graphite battery paper in the same ionic liquid in Chem. Comm. with a submission date of 27th January 2015 and acceptance date of 15th June 2015 (Chem. Comm., 51, 11892-11895). The paper did not cite our work. The title of their Chem. Comm. paper was "A new aluminium-ion battery with high voltage, high safety and low cost".

So in terms of journal publications: the fact is Jiao's manuscript was submitted 8 months after our Science manuscript was rejected, and it was published 2 months after our Nature paper was published without citing our paper.

The paper mentioned by the reviewer (Journal of The Electrochemical Society, 160 (10) A1781-A1784 (2013)) by Rani et al. from India entitled "Fluorinated Natural Graphite Cathode for Rechargeable Ionic Liquid Based Aluminum-Ion Battery" was already cited in our Nature paper as Ref.6. The Al/fluorinated graphite battery was only charged to 1.2 V and exhibited a capacitive discharge behavior without redox related plateaus and gave much lower discharging voltages ~ 1 V, which differed greatly from the characteristics of our batteries with an energy storage mechanism clearly different from the chloroaluminate anion intercalation/de-intercalation redox mechanism in the

high quality graphite in our battery. In our revision of the current manuscript, we have clearly stated this difference in the introduction.

3. Energy Density of Al ion battery

In our manuscript, the current Al-graphite battery using high quality graphite produces an energy density of $\sim 68.7 \text{ Wh kg}^{-1}$ (based on $\sim 110 \text{ mAh g}^{-1}$ cathode capacity and the masses of active materials in electrodes and electrolyte). Ideally, the energy capacity of Al-graphite battery could reach 40 Wh/kg suggested by the reviewer. This will require engineering efforts. Another direction is to lower the cost of ionic liquids so that the energy/cost of the Al battery becomes highly competitive over Pb acid battery. We have already made some progress along this line in the past year.